# Coaching styles and sports motivation in athletes with and without Intellectual Impairments

**Kandianos Emmanouil Sakalidis, Florentina Johanna Hettinga◯\*, Fiona Chun Man Ling**

Faculty of Health and Life Sciences, Department of Sport Exercise and Rehabilitation, Northumbria University, Newcastle upon Tyne, United Kingdom

\* florentina.hettinga@northumbria.ac.uk

## Abstract

The cognitive limitations of athletes with Intellectual Impairments (II) may influence their sport behaviour and lead them to rely on coaches' support. However, it is still unclear how II may influence sports performance progression and motivation and how coaches perceive their athletes with II and coach them. Thus, this study aims to examine 1) coach's perceptions of motivation and performance progression in athletes with and without II, 2) coaching style (dis)similarities, and 3) the association between these factors. Coaches of athletes with ($n = 122$) and without II ($n = 144$) were recruited and completed three online questionnaires, analysed using a series of non-parametric analyses ($p \leq .05$). Results showed that perceived performance progression and controlled motivation were higher of athletes with II while perceived autonomous motivation was higher of athletes without II. No coaching style differences were found between the two groups. Additionally, a need-supportive coaching style negatively predicted amotivation, and a need-thwarting coaching style predicted lower autonomous motivation in athletes with II only. Overall, it seems that the coaches perceived that their athletes with II demonstrate different motivations and react dissimilarly to their coaching styles compared to athletes without II. They may also adopt different standards of sporting success for them. Due to these differences, it is important to offer appropriate training and knowledge to coaches about disability sports and the adaptations needed to effectively coach athletes with II. In summary, this paper gives some insights about the coach-athlete relationship and highlights the necessity to further support the sports development of people with II.

**Data Availability Statement:** We will store data in the University repository: https://library.northumbria.ac.uk/research-data-management/figshare This is the doi:https://doi.org/10.25398/rd.northumbria.23284472.v1.

## Introduction

According to the convention of the rights of individuals with disabilities, people with Intellectual Impairments (II) have the right to participate in the sports activity of their choice [1]. Previous studies have highlighted the importance of engagement in sports for people with II [2, 3]. Sports participation can improve cardiovascular endurance, muscle strength and motor skills [2], and it can enhance psychological well-being and cognitive skills improvement of this

**Funding:** The author(s) received no specific funding for this work.

**Competing interests:** The authors have declared that no competing interests exist.

population [2], [3]. It can also facilitate the development of athletes' transferable skills, like the ability to follow instructions and complete independent tasks [4]. Physical fitness and sports-related skills improvements serve as mediators for increased motivation in people with II [5]. However, only a limited number of people with II regularly participate in recreational or competitive sports, compared to people without II [6]. Due to the insufficient levels of sports participation and the additional health issues, their general fitness is significantly lower compared to the average population [7, 8].

Moreover, the intellectual functioning (IQ≤70) and adaptive behaviour deficits in people with II [9] can negatively impact physical, physiological, psychological, and social aspects of their sports performance [10–13]. For instance, skills like self-regulation, decision-making, and learning by experience, which are important in sports performance and proficiency, are underdeveloped in persons with II [10, 14]. Moreover, due to the impaired reasoning and judging abilities, athletes with II could misinterpret the others' social behaviour (e.g., coaches, teammates and/or opponents) and respond differently to the environmental cues [15]. In running trials for example, the performance feedback that the social environment offers (e.g., coaches) cannot facilitate the ability of people with II to maintain a steady pace [13], a critical skill for optimal performance progression [16, 17].

A theoretical framework that can explain the influence of coaches' attitudes on athletes' motivation, self-regulatory behaviour and sports performance progression, is Self-Determination Theory (SDT) [18, 19]. SDT is a macro-theory of human motivation that makes a distinction between autonomous (e.g., individuals identify an activity as valuable or personally meaningful) and controlled motivation (e.g., individuals engage in an activity for external reasons) [18]. Autonomous motivation includes three types of motivation regulation–intrinsic regulation, integrated regulation and identified regulation. Controlled motivation includes two types of motivation regulation–introject regulation and external regulation. Both autonomous and controlled motivation direct behaviour, unlike amotivation, which refers to a lack of motivation [18]. According to SDT, motivation orientation depends on the satisfaction of three psychological needs [18]. Psychological needs refer to the inherent need for competence (e.g., through mastering an activity, positive reinforcement, winning a competition), autonomy (experience of volition) and relatedness (social environment's support) [18]. These needs are critical in athletes with II as they guide their sports behaviour and facilitate their long-term engagement in sports [20]. However, the cognitive impairments (e.g., reasoning and judging), the high anxiety levels, and the low self-esteem of people with II [9, 21], could negatively influence their autonomous motivation and in turn, hinder their sports performance progression [12, 19, 22].

In sports settings, coaches create a context through which their coaching style can support (need-supportive) or thwart (need-thwarting) athletes' fulfilment of psychological needs [18, 23]. On the one hand, need-supportive coaches can promote athletes' autonomous motivation (more self-determined behaviour). This type of motivation has a significant impact on athletes' long-term sports participation and performance progression, as it is associated with better learning, effort, and persistence [19, 24]. On the other hand, need-thwarting coaches can promote athletes' controlled motivation (externally regulated behaviour), which is considered less optimal as it is related to negative outcomes like burn-out and failure [18, 24, 25]. Due to the cognitive deficits of athletes with II (e.g., self-regulation and decision-making), this population tends to be more reliant on others [26] and could subsequently lead their coaches to adopt a more need-supportive coaching style. Moreover, these cognitive deficits could lead athletes with II to judge and respond differently to their coaches' coaching styles compared to athletes without II [11, 15].

In an effort to ensure an inclusive and fair sporting system, there has been a growing emphasis on mainstreaming disability sports in recent years [27]. Mainstreaming aims to integrate disability and non-disability sports organisations and to offer a range of possible and inclusive sports and exercise opportunities to people with disabilities [28]. Athletes of all abilities and their coaches play a central role in supporting the mainstreaming development. Therefore, to offer appropriate inclusive sports environments to people with II, it is imperative to understand more about their sports performance progression and motivations as well as the coach-athlete relationship. Additionally, a better knowledge about the differences of the aforementioned variables between people with and without II could facilitate a smoother mainstreaming in sports and offer more exercise pathways to people with II [27]. However, as athletes with II are one of the most understudied populations in sports settings, it is not well-documented how to properly include them in sports and how to guide coaches during this process [29]. Moreover, even if it is evident that coaches could affect athletes' motivation and sports performance development, especially for athletes without II [23–25], it is not yet clear the impact of II on sports performance progression and motivation. Moreover, it is still unknown the role of coaches towards athletes with II and how this might differ compared to athletes without II. In this study, we chose to focus on coaches' reports because by exploring coaches' perceptions of their athletes' performance progression and motivation, we can better examine the relationships between these perceptions and their coaching styles. We can also explore how coaches' perceptions can promote (or restrict) the inclusion of people with II in sports [30, 31]. The researchers are aware of the lurking danger of promoting intellectual ableism when a proxy respondent is preferred over a person with II [32] thus, their future studies aim to 'give a voice to the voiceless'. For this study however, the exploration of the athletes' motivation and progression from different perspectives will give us the opportunity to explore the coach-athlete relationship in sports settings more deeply [30].

Therefore, this study is based on the theoretical framework of SDT, and it aims to examine if: 1) there are differences in sports performance progression and motivation orientations between athletes with and without II as reported and perceived by their coaches, 2) there are differences in coaching styles between coaches of athletes with and without II, and 3) coaching styles are predictors of sports performance progression and motivation orientation in athletes with and without II. We hypothesize that: 1) coaches of athletes with II perceive their athletes to have made less progression in their sports performance and have adopted more controlled types of motivation compared to athletes without II, 2) coaches of athletes with II will adopt a need-supportive coaching style, compared to coaches of athletes without II, and 3) coaching styles are predictors of sports performance progression and motivation orientation in athletes with II and these predictors differ between the two groups (II and non-II).

## Materials and methods

### Participants and recruitment

Recruitment of coaches of athletes with and without II was done through sports organisations, recreational centres and sports clubs via phone calls and e-mails (from January until May of 2021). The authors did not have access to information that could identify individual participants during or after data collection. Two hundred and sixty-six coaches with coaching experience in different sports (e.g., athletics, gymnastics, basketball, football etc.) consented to participate (45.9% coaches of athletes with II). Coaches' average age was 40 ($SD$ = 16, range 17 to 81 years old) and 58.6% of them were male. The mean coaching experience for coaches working with athletes with II was 11 years ($SD$ = 10) while coaches of athletes without II had a mean average coaching experience of 15 years ($SD$ = 12). Both groups of coaches had

experience coaching a variety of individual and team sports, like fencing, boccia, archery, athletics, football, and basketball. We included coaches who were fluent in English, had at least one year of coaching experience, and whose athletes were adolescents or adults (aged 12 or above) with or without II. Coaches were asked to act as proxy respondent for a group of persons (athletes with or athletes without II) [33] and provide their overall view of their athletes' motivation, similarly to Rocchi [34]. For comparability purposes, their athletes with or without II were categorized into the 'participation' or 'performance' stage of sports development (focus on sports skills development with experience in local or regional, recreational competitive events) [35]. Athletes with II must meet the criteria for diagnosis of II as set by the British Psychological Society [9]: limitations in intellectual and adaptive functioning with an IQ ≤ 70, limitations in social, practical, and conceptual skills, and manifested before the age of 18 years. The study was reviewed and approved by the Institutional Ethics Board.

## Materials and methods

Coaches of athletes with and without II completed questionnaires via an online platform (JISC). All coaches completed the 3 questionnaires overviewed below, which lasted approximately 20 minutes.

**Rated performance (coaches' reports of athletes' sports performance progression).** This instrument is completed by coaches and is used to investigate the extent to which the athletes had progressed in the (a) physical, (b) tactical, (c) technical, and (d) psychological domain over the past year [36]. As this is an objective measurement of athletes' perceived sports performance progression, the rate of progression was based on the sports performance abilities of the group of athletes and how coaches perceived their expected rate of improvement. These items are combined and form an intraindividual athletic performance scale (total performance progression). The scale uses a 7-point scale, ranging from 1 (strong regression) to 7 (strong progression) and showed excellent internal consistency [36]. Due to the lack of exercise and training routine during the COVID-19 outbreak, coaches were instructed to complete the questions retrospectively.

**Revised sports motivation scale—perceived player motivation.** This instrument is founded on the SDT [18] and assesses coaches' perspectives of athletes' reasons for participating in sports (e.g., 'because they feel better about themselves when they do play'; 'because people around them reward them when they do play') [34, 37]. The scale measures sports motivation according to six types of behavioural regulation—intrinsic regulation, integrated regulation, identified regulation (under the autonomous motivation subscale; 9 items), introjected regulation, external regulation (under the controlled motivation subscale; 6 items), and amotivation (3 items). The scale uses a 7-point scale, ranging from 1 (does not correspond at all) to 7 (corresponds completely). This instrument showed a strong factor structure and acceptable internal consistency [34].

**Interpersonal behaviours questionnaire—self (IBQ-self).** This questionnaire is also founded on the SDT [18] and assesses coaches' reports of their own interpersonal behaviours (IBQ-self) in sports settings [38]. The questionnaire consists of 24 items (e.g., 'when I am with my athletes, I provide valuable feedback'; 'when I am with my athletes. I pressure them to adopt certain behaviours') and six subscales—autonomy-supportive, competence-supportive, relatedness-supportive (collectively they form the need-supportive scale), and autonomy-thwarting, competence-thwarting, and relatedness-thwarting (collectively they form the need-thwarting scale). The measure uses a 7-point scale, ranging from 1 (do not agree at all) to 7 (completely agree) and showed a strong factor structure, internal consistency, and validity [38].

## Statistical analysis

Perusal of the data using the Shapiro-Wilk test of normality suggests that the assumption of normality is violated. Therefore, we conducted a series of non-parametric analyses. To address aims 1 and 2 (e.g., differences in perceived total performance progression, perceived motivation orientation and coaching styles), we conducted a rank MANOVA to test if there were differences in autonomous motivation, controlled motivation, amotivation, total performance progression, need-supportive, and need-thwarting differences (six dependent variables) between the reports of coaches of athletes with and coaches of athletes without II (group; independent variable).

To address aim 3 (e.g., predictors of total performance progression and motivation orientation), we first performed two Spearman correlation analyses to assess the relationship between the variables for each group. Variables indicating significant correlations with coaching styles were entered into a series of Additive Nonparametric Regressions (Generalized additive model), with need-supportive and need-thwarting coaching styles as independent variables. The statistical analyses were performed using $R$, version 4.1.1, and the level of significance was set at $p \leq .05$.

## Results

The rank MANOVA analysis showed that there were no group differences between coaches' need-supportive ($p = .53$) and need-thwarting style ($p = .41$) and no group differences between perceived athletes' amotivation ($p = .63$). Furthermore, perceived autonomous motivation was significantly lower ($p < .001$) and perceived controlled motivation was significantly higher in athletes with II compared to athletes without II ($p < .001$). Finally, perceived total performance progression of athletes with II was significantly higher compared to athletes without II ($p = .01$) (see Table 1 for the descriptive data, and Fig 1 for the univariate post-hoc comparisons between the variables).

Results of the Spearman Correlation analyses indicated that both coaching styles (need-supportive and need-thwarting) were significantly correlated with autonomous motivation and amotivation in II and non-II group ($p < .001$). Additionally, both coaching styles were significantly correlated with the total performance progression in non-II group ($p < .001$) (see

**Table 1. Descriptive statistics of need-supportive, need-thwarting, performance progression, autonomous motivation, controlled motivation and amotivation for athletes with and without II.**

| Variable | Source | N | Mdn (IQR) | M (SD) |
|---|---|---|---|---|
| Need-Supportive | II | 122 | 68.00 (17.00) | 66.02 (11.59) |
| | Non-II | 144 | 71.00 (19.50) | 66.36 (12.67) |
| Need-Thwarting | II | 122 | 31.00 (21.00) | 32.95 (12.65) |
| | Non-II | 144 | 31.50 (26.50) | 35.56 (16.34) |
| Performance Progression | II | 122 | 22.00 (5.00) | 21.19 (4.21) |
| | Non-II | 144 | 21.00 (6.00) | 19.82 (4.38) |
| Autonomous Motivation | II | 122 | 44.00 (15.00) | 43.75 (9.72) |
| | Non-II | 144 | 51.00 (12.50) | 49.83 (9.91) |
| Controlled Motivation | II | 122 | 29.00 (8.00) | 29.34 (5.93) |
| | Non-II | 144 | 25.00 (7.50) | 25.40 (6.08) |
| Amotivation | II | 122 | 8.00 (6.00) | 8.36 (4.36) |
| | Non-II | 144 | 8.00 (6.50) | 8.63 (4.48) |

N = number of items (coaches' reports). Mdn = Median, IQR = Interquartile Range, M = mean. SD = standard deviation.

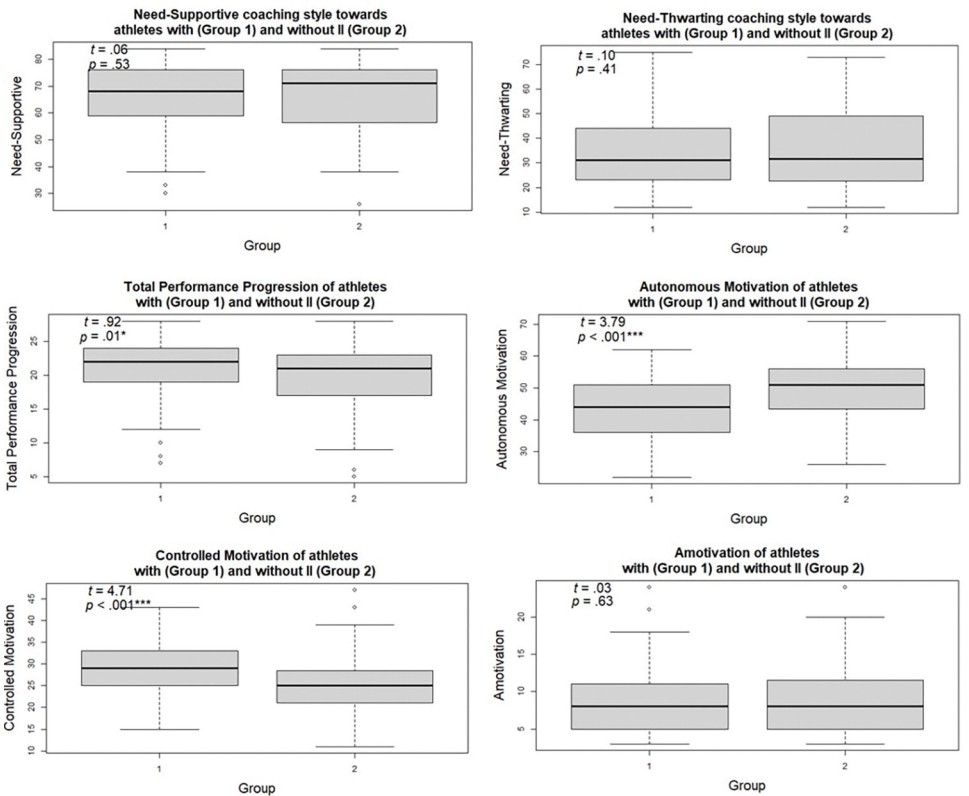

**Fig 1. Results of the rank MANOVA.** Univariate post-hoc comparisons of Need-Supportive, Need-Thwarting, Total Performance Progression, Autonomous Motivation, Controlled Motivation and Amotivation for athletes with and without II. T = T Value, P = P Value (* shows the mean differences are significant at the .05 level; ** shows the mean differences are significant at the .01 level; *** shows the mean differences are significant at the .001 level).

Table 2). Therefore, only these variables were entered into the series of Additive Nonparametric Regression analyses.

A series of Additive Nonparametric Regressions were run to examine if need-supportive and need-thwarting were predictors of autonomous motivation and amotivation in the II group. A second series of Additive Nonparametric Regressions were run to examine if need-supportive and need-thwarting were predictors of autonomous motivation, amotivation and total performance progression in non-II group. Results showed that a need-supportive coaching style positively predicted autonomous motivation in athletes with and without II ($p < .001$, *adj*. $R^2 = .28$ and $p = .00$, *adj*. $R^2 = .47$ respectively). It also negatively predicted amotivation in athletes with II ($p = .00$, *adj*. $R^2 = .25$). Additionally, a need-thwarting coaching style positively predicted amotivation in athletes with and without II ($p = .02$, *adj*. $R^2 = .25$ and $p < .001$, *adj*. $R^2 = .37$ respectively), and negatively predicted autonomous motivation in athletes without II ($p = .00$, *adj*. $R^2 = .47$). Fig 2 presents the partial effects plots with the approximate significance of smooth terms for predictors of autonomous motivation and amotivation in athletes with and without II. Neither coaching style significantly predicted total performance progression in both groups.

## Discussion

This study aimed to shed light on athletes' sports performance progression, athletes' motivation orientations and coaching styles differences, as well as the relationships between these

**Table 2. Spearman correlation matrix for need-supportive, need-thwarting, total performance progression, autonomous motivation, controlled motivation and amotivation for -athletes with and without II.**

| Group | Variable | 1 | 2 | 3 | 4 | 5 | 6 |
|---|---|---|---|---|---|---|---|
| II | | | | | | | |
| | 1. Need-Supportive | - | | | | | |
| | 2. Need-Thwarting | -.62** | - | | | | |
| | 3. Total Performance Progression | .13 | -.08 | - | | | |
| | 4. Autonomous Motivation | .51** | -.45** | -.01 | - | | |
| | 5. Controlled Motivation | -.06 | .03 | .13 | -.19* | - | |
| | 6. Amotivation | -.45** | .41** | -.24** | -.41** | .05 | - |
| Non-II | | | | | | | |
| | 1. Need-Supportive | - | | | | | |
| | 2. Need-Thwarting | -.82** | - | | | | |
| | 3. Total Performance Progression | -.19* | .24** | - | | | |
| | 4. Autonomous Motivation | .65** | -.57** | .04 | - | | |
| | 5. Controlled Motivation | -.00 | .10 | .01 | -.07 | - | |
| | 6. Amotivation | -.48** | .62** | .10 | -.45** | .22** | - |

\* Shows the mean differences are significant at the p < .05 level

\*\* shows the mean differences are significant at the p < .01 level.

factors, as reported and perceived by coaches of athletes with and without II. The results did not fully support our first hypothesis that coaches of athletes with II perceive their athletes to have made less progress in their sports performance and have adopted more controlled types of motivation compared to athletes without II. More specifically, coaches' reports indicated that athletes with II were perceived to progress more in their sports performance (total performance progression) compared to athletes without II. However, as we hypothesised, coaches reported and perceived that athletes with II adopt more controlled types of motivation than athletes without II, and less autonomous types of motivation, than athletes without II.

A reason that total performance progression of athletes with II is perceived to be higher could be due to lower long-term engagement in sports and the lower levels of physical fitness and muscle strength of this population compared to athletes without II that previous studies reported [39, 40]. However, due to the nature of the Rated Performance questionnaire (was based on the perceived physical, tactical, technical, and psychological progression of the athletes) and the plethora of different sports that coaches were coaching, we approach this argument with caution. More research based on objective measurements is needed to explore the relationships between training age and physical fitness (e.g., fitness assessments that test the strength and muscle mass alternations of athletes) with the sports performance progression of athletes with II [41, 42].

However, as this study was based on coaches' perceptions, a more appropriate explanation for these findings could be the disability stereotype where achievements by people with disabilities are rated more positively from the able-bodied society [43]. Thus, coaches of athletes with II might unconsciously adopt different standards for sporting success and overestimate their total performance progression [44]. For instance, coaches may have relatively low expectations from their athletes with II, while a great physical, tactical, technical, and/or psychological progression of them could be perceived by the coaches as paradoxical [38]. Additionally, coaches of athletes with II tend to adopt a mentorship role, focus less on their athletes' sports performance development, and potentially underestimate the importance of nurturing the athletic identity that athletes with II may wish to develop [45]. These attitudes may be well-intentioned however, when

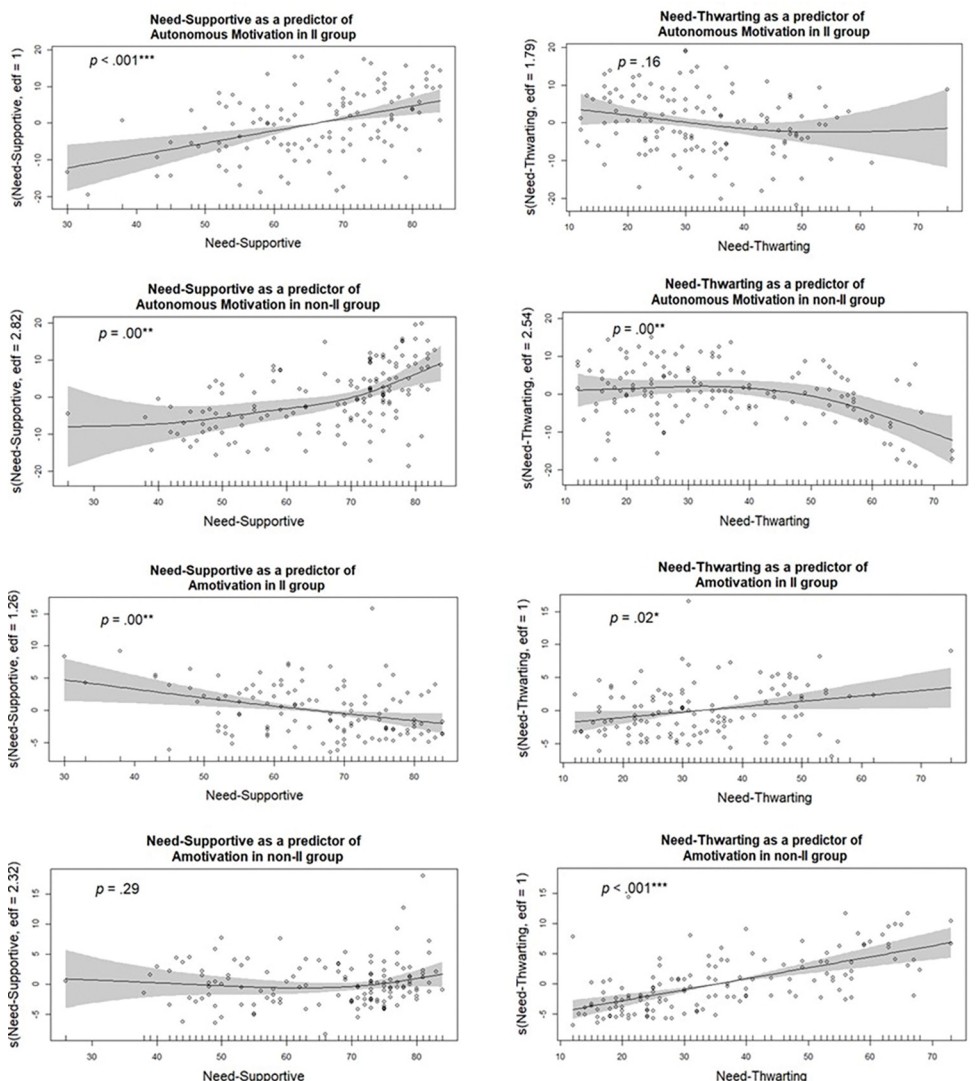

**Fig 2. Partial effects plots with partial residuals and standard errors (95% confidence interval) and approximate significance of smooth terms for predictors of autonomous motivation and amotivation in athletes with and without II.** Edf = effective degrees of freedom, P = P Value (* shows the mean differences are significant at the .05 level; ** shows the mean differences are significant at the .01 level; *** shows the mean differences are significant at the .001 level).

athletes with II accomplishments are portrayed as surprising and/or inspirational it can perpetuate ableism [38, 44]. If these unintentional (but still ableist) attitudes occur, this could make the mainstreaming of disability sports more challenging for this population. Thus, there is a necessity to reshape coaches' assumptions of what athletes with II can and cannot do and help them set realistic sports performance goals for their athletes. Moreover, even if we tried to recruit coaches who are working with athletes with a similar stage of sports development, we recognise that disability and non-disability sports organisations are not fully integrated [27]. As a result, the training sessions, the opportunities for sports performance development, and participation in competitive events may vary for athletes with and without II [27]. Consequently, the coaches' expectations regarding their athletes' improvement may also differ between the two groups and could partially explain the findings observed in this study.

The results also indicated that athletes with II adopt more controlled types of motivation compared to athletes without II and less autonomous types of motivation (as perceived by their coaches). Previous research has shown that athletes with II exhibit higher ego orientation and lower self-regulation compared to athletes without II [10, 46]. This could partly explain the lower levels of long-term participation in sports of athletes with II compared to athletes without II [47, 48], as autonomous motivation functions in a dyadic relationship with self-regulation and facilitates athletes' long-term exercise engagement and persistent sports behaviour compared to athletes who adopt more controlled forms of motivation [18, 24, 25]. The higher level of perceived controlled motivation could be a result of the high levels of anxiety, decreased confidence and social phobia experienced by people with II and may influence their sports motivation [49, 50]. In addition, the lack of awareness and societal support that athletes with II reported [51], could hinder the fulfilment of their relatedness' needs [18] which could, in turn, fuel more controlled types of motivation compared to athletes without II. However, the motivational differences between athletes with and without II could have occurred due to the difficulties of proxies (such as coaches) to recognise that people with II can have a good, personally meaningful life [32, 43, 44] and accept the role of people with II in their own autonomous decision-making [32]. Coaches in sports settings tend to prioritise their own aspirations and perspectives regarding the needs of people with II, potentially overshadowing their athletes' sports motivations [45]. In addition, coaches may observe that the social environment (e.g., parents) hinders the decision-making of people with II and consequently adopt an overprotective stance towards them [45]. Thus, coaches may perceive that the sports participation of athletes with II depends more on external and less on internal motivations compared to athletes without II, but further research is needed to explore the level of intellectual ableism in coaching settings and give equal attention to both athletes with II and their coaches [32].

The results of our study did not support our second hypothesis, indicating that the coaching style between the two groups is similar. Given the coaching experience of the participants, with both groups having an average coaching experience of over 10 years, it is unlikely that the observed similarities in coaching styles can be attributed to a lack of experience or their experience differences. A possible explanation for the coaching style similarities could be that most of the coaches of athletes with II come from mainstream sports and have a traditional coaching education background [52]. Previous studies in sports for people without II showed that coaching behaviour is influenced by athletes' motivation [53, 54]. However, the different motivation orientations of athletes with and without II and the similar coaching styles of their coaches, indicate that coaching behaviour towards athletes with II seems less adapted to athletes' motivation. Moreover, these findings could indicate that coaches may have difficulties in adapting their approach to the needs of athletes with II; thus, more effort is needed to enhance the autonomous motivation of this population. Due to the reciprocal relationship between coaching behaviour and athletes' motivation [54], future qualitative research should further investigate the coach-athlete relationship in II sports, coaches' practices, how and why they implement them, and how beneficial this could be for their athletes' long-term sports participation and development.

The series of additive nonparametric regression analyses partially supported our third hypothesis, indicating that coaching styles are predictors of motivation orientation in athletes with II and that these predictors differ between the two groups (II and non-II). Specifically, the results show that the coaches' need-supportive style is a predictor of the autonomous motivation (positive) and amotivation (negative) of athletes with II. At the same time, coaches' need-thwarting style positively predicts amotivation in this population (Fig 2). These findings indicate the importance of the coach-athlete relationship in II sports and suggest that athletes with II may have the capability to respond accordingly to different coaching styles contrary to

common beliefs [15]. For example, athletes with II may feel a sense of ownership and enjoyment as well as reduced feelings of disinterest when their coaches take time to understand their feelings and needs and provide them with choices and encouragement (need-supportive coaching style) [23, 25]. Additionally, they may feel disengaged, demotivated, and uninterested in participating in sports when their coaches tend to be controlling or neglectful of their needs (need-thwarting coaching style) [23].

The findings also highlight the necessity of coaches to nurture the basic psychological needs of athletes with II. Coaches of athletes with II may wish to provide their athletes with choices and meaningful rationales for the assigned exercises and show trust in their capabilities regardless of their cognitive limitations. They could also consider giving them clear and simplified instructions and providing them the opportunity to express their needs and anxieties in a socially safe and supportive sports environment [24]. These attitudes may be essential for athletes with II, a population dealing with increased anxiety, social phobia, and decreased confidence [49] and tending to adopt more controlled types of motivation. Coaches' need-supportive style may increase athletes' chances to adopt more autonomous motivation regulations, avoid amotivation, increase their positive affect [23], facilitate their self-regulatory development and inspire their long-term engagement in sports [24]. Contrariwise, coaches who thwart athletes' basic psychological needs could engender feelings of pressure, failure, and loneliness [23], demotivate them from continued sports participation (amotivation), and increase their chances of depression and burnout [23]. Thus, it is optimistic that coaches of athletes with II are trying to connect with their athletes, promote the social interaction, and focus on their athletes' positive emotions in sports settings. However, more research is needed to assess the efficiency of their approach and how they can promote a fertile ground for their athletes' long-term engagement in sports [45].

It is also important to explore the different role that the coaching styles have in athletes with and without II. It seems that the need-supportive style predicts autonomous motivation only in athletes with II. On the other hand, the need-thwarting style predicts amotivation only in athletes without II. The impaired cognitive abilities of athletes with II could lead them to respond differently to environmental cues (e.g., coaches' attitude) and react dissimilarly to coaching styles compared to athletes without II [9, 11, 13]. These differences, along with the different motivation orientations between athletes with and without II, should be taken into consideration in future sports disability education programs. It is thus crucial to educate coaches of athletes with II on how to effectively deal with the cognitive deficits of this population, interact with them appropriately, and provide effective support for their basic psychological needs [52, 55]. Nonetheless, education for coaches regarding disabilities will be beneficial and will facilitate the mainstreaming development in sports only if the coaches acknowledge that each athlete (II and non-II) has a unique personality, and that they should adapt their behaviour to each athlete's needs in order to foster meaningful athlete-coach relationships [56, 57]. Currently, coaching education opportunities within disability sports are still lacking, which makes it even more challenging for coaches to gain any advanced learning about the most progressive and effective ways to coach athletes with II and offer them inclusive sports opportunities [52].

This study presented some limitations that need to be addressed. First, due to the lack of validated self-report instruments that measure motivation orientations in athletes with II, this study was based on coaches' perception of athletes' motivation orientations. However, the communication difficulties that athletes with II experience could lead their coaches to misinterpret their needs, behaviours, and motives [58]. Future studies should investigate the role of significant others (e.g., coaches, carers, parents, peers) in fostering different motivation orientations. Future research should also aim to develop appropriate and valid instruments that

measure motivational regulations among athletes with II. Another limitation of the study is the absence of qualitative feedback in the survey data. Qualitative approaches, like interviews with coaches and athletes with II, could provide a deeper understanding of the coach-athlete relationship in disability sports and capture nuanced information that our quantitative approach alone may not revealed [59]. Thus, future studies could use a mixed-methods approach (combining qualitative and quantitative methods) to obtain a more comprehensive picture of the problem [59]. Future studies should also ensure the active involvement of participants with II and their contribution to the research process. Additionally, it is crucial for these studies to also consider other relevant stakeholders (e.g., family members, support staff, policy makers) in examining the coach-athlete relationship in disability sports and the inclusive practices towards athletes with II [45]. Another limitation is also the lack of device-based measurements that investigate the sports performance progression of athletes. A criticism of self-report measurements of sports performance development is that they could be affected by coaches' bias towards athletes who have specific roles within the team [60]. However, due to COVID-19 restrictions during the data collection process, it was not feasible to include device-based measurements of sports performance. Future work could integrate both device-based and self-report performance assessments to gain a better understanding of athletes' progression and better support their long-term development in sports performance settings [60].

## Conclusion

In summary, this paper gives some insights about the significance of the coach-athlete relationship in sports and the importance of a need-supportive coaching style to enhance autonomous motivation and prevent the amotivation of athletes with II. While self-reported coaching styles were similar between coaches of athletes with and coaches of athletes without II, their perceptions of their athletes' performance progression and motivation orientations seemed to differ. This might have occurred due to the differences in sports opportunities and experiences between athletes with and without II and/or due to the different sports standards that their coaches adopt. Thus, it is important to offer appropriate training and knowledge to coaches about disability sports and the adaptations needed to effectively coach athletes with II and to appropriately offer them inclusive sports activities.

## Supporting information

**S1 Checklist. STROBE statement—checklist of items that should be included in reports of observational studies.**
(DOCX)

**S2 Checklist.** *PLOS ONE* **clinical studies checklist.**
(DOCX)

## Author Contributions

**Conceptualization:** Kandianos Emmanouil Sakalidis, Florentina Johanna Hettinga, Fiona Chun Man Ling.

**Formal analysis:** Kandianos Emmanouil Sakalidis.

**Investigation:** Kandianos Emmanouil Sakalidis.

**Methodology:** Kandianos Emmanouil Sakalidis, Florentina Johanna Hettinga, Fiona Chun Man Ling.

**Supervision:** Florentina Johanna Hettinga, Fiona Chun Man Ling.

**Writing – original draft:** Kandianos Emmanouil Sakalidis.

**Writing – review & editing:** Florentina Johanna Hettinga, Fiona Chun Man Ling.

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
