## [Decision Letter · Decision Letter 0]

3 May 2023

PONE-D-22-31733Coaching styles and sports motivation in athletes with and without Intellectual ImpairmentsPLOS ONE

Dear Dr. Hettinga,

Thank you for submitting your manuscript to PLOS ONE. After careful consideration, we feel that it has merit but does not fully meet PLOS ONE’s publication criteria as it currently stands. Therefore, we invite you to submit a revised version of the manuscript that addresses the points raised during the review process.

We look forward to receiving your revised manuscript.

Kind regards,

Ali B. Mahmoud, Ph.D.

Academic Editor

PLOS ONE

Journal Requirements:

3. Please ensure that you include a title page within your main document. You should list all authors and all affiliations as per our author instructions and clearly indicate the corresponding author.

4. Please remove your figures from within your manuscript file, leaving only the individual TIFF/EPS image files, uploaded separately. These will be automatically included in the reviewers’ PDF.

Reviewers' comments:

Reviewer's Responses to Questions

**Comments to the Author**

1. Is the manuscript technically sound, and do the data support the conclusions?

Reviewer #1: Yes

Reviewer #2: Yes

2. Has the statistical analysis been performed appropriately and rigorously? 

Reviewer #1: Yes

Reviewer #2: Yes

3. Have the authors made all data underlying the findings in their manuscript fully available?

Reviewer #1: No

Reviewer #2: Yes

4. Is the manuscript presented in an intelligible fashion and written in standard English?

Reviewer #1: Yes

Reviewer #2: Yes

5. Review Comments to the Author

Reviewer #1: I would like to congratulate the authors on an important piece of work. This manuscript highlights the need for further work in this area and because of that, most of my critical thoughts below are out loud thinking to what is really missing in the literature that can further inform some of the findings here. I recognize the importance of capturing a wide range of participants via the survey mechanism and the importance this brings to the literature, however, the shortcomings of this approach are the lack of clarification/conclusions we can draw based on the quantitative data. Tied to that, my other concern is the lack of athlete-voice. While the authors address this front and center at the beginning of the manuscript, I recommend them to be mindful of this, throughout the discussion where conclusions are drawn.

Most my comments are specific to the text, however, I do urge the authors to revisit the discussion section to see whether some of the conclusive remarks made can be reconsidered given the limited information we have with regards to this cohorts’ development and the need to further unpack some of the nuances here before we can make such conclusions.

Line 89: close bracket

Line 101: hinder or impact their progression rate? Reword here, ends awkwardly

Line 136: replace trying with ‘aims’ - this is an interesting point you bring up as I was curious to the significance of capturing the athlete/family perspective of this environment, and their experiences

Line 207: did any of the coaches coach both II and non-II athletes? To this, it might be helpful to get a bit more description of the coaches and sports involved, i.e., years coaching, highest level coaches, level of expertise of athletes they coached (that were rated), sports involved, and cluster of sports, etc.

Line 254-256: I wrestle with this idea and concerned about how it is being interpreted. Isnt rate of progression relative to the athletes’ ability and the coaches’ expectation of their rate of improvement based on their baseline vs growth over time to reach their personal potential? So how is the rate of improvement being measured between the athletes? Did the question ask for rate of improvement relative to the athletes’ expected improvement over that time? And did the athletes’ impairment influence the coaches’ initial expectation standards in any way?

Tied to this, I think relevance of athletes’ level of competition and understanding how that distribution lies among the II and non-II becomes more relevant

Line 259-265: hard to make this conclusion given the question wasn’t asked of coaches on what they rated as ‘improvement’ and what the metrics were that they considered to analyze athletes’ performance – especially given a range of sports that were involved, ie table-tennis requires a technical proficiency that is more prominent in assessing one’s rate of improvement relative to a cycling where post-balancing, the most relevant measure of improvement is cardiovascular output

Line 269-273: more positively and/or the bar is set lower to start with (knowing how many athletes with an impairment the coach has worked with previously would help narrow this assumption down slightly, but these are challenges to surveys vs qualitative studies, harder to draw on coaches’ observations or perception through quantitative measures, so I’d be cautious of the extent that conclusions are drawn from the findings)

Line 298: and also more research unpacking this directly with the athletes whom has II

Line 300-302: more context/info on coaches’ experience would help reader make assumptions/draw conclusions on this and other relevant topics

Line 320-321: Athletes perhaps respond to the environment that the coach facilitates based on their methods, i.e., they might not directly draw comparisons but experience sense of belongingness, etc. I find this conclusion challenging, as it sounds like athletes explicitly need to deduce coaches’ methods and their preferential against that type of coaching? What if the athlete had never experiences other types of coaching? Is it common for athletes to explicitly identify the coaches’ methods?

Line 322-334: I suggest rewording of some recommendations here, ‘should’ comes off a bit too strong? Especially given how much more work is needed in this area both qualitatively and quantitatively before we make assertions, especially given this was a survey done with coaches and athletes were not directly involved?

Limitations: lack of qualitative feedback to the survey info + athlete input is very important acknowledgement, some of the elements you speak of here highlight the need to introduce qualitative work into the quantitative data, highlighting the importance of taking a mixed-methods approach while incorporating the breadth of stakeholders involved in the DTE that impact athletes’ development (i.e., parents, teammates, directors, support staff, coaches, and most importantly, athletes themselves).

Reviewer #2: REVIEW:

The article presented is of a relevant topic and its scientific structure is correct. It addresses a topic of interest in sports science and physical activity for people with disabilities. The vision and study of a sports training process through the coaches' perspective is coherent and the methods and instruments used in the research are appropriate for a scientific article of this level.

The results, discussion and conclusions are in line with the hypotheses and objectives proposed under an updated theoretical and bibliographical framework and with a good level of academic impact.

However, in the review of the article, it presents a series of elements that in the opinion of this reviewer should be improved/adapted.

In the introduction, in line 99 in this paragraph, the statement "could negatively influence their autonomous motivation and in turn, make them progress less in sports" is only supported by a single article, which is true that it was carried out with a good sample, but it would be important to strengthen this argument with other publications.

In the materials and methods in line 159 in relation to the selection of the sample, the criterion of only one year of experience does not seem to be an indicator of being a factor that conveys through your answers a fully founded opinion on the questionnaires that you propose in the study, the lack of experience may be an element that subtracts accuracy from the answers by not being supported by practical knowledge developed through experience.

In the discussion line 264 when you talk about "Thus, our speculation is that the total performance progression of athletes with II might be more obvious to their coaches." I think this part needs more argumentation.

In line 292 on the reflection "However, the motivational differences between athletes with and without II could have occurred due to the difficulties of proxies (such as coaches) to recognise that people with II can have a good, personally meaningful life [38] and accept the role of people with II in their own autonomous decision-making [32]." It is not clear that the quote [38] is well used here, perhaps I should argue this paragraph more.

6. PLOS authors have the option to publish the peer review history of their article (what does this mean?). If published, this will include your full peer review and any attached files.

Reviewer #1: No

Reviewer #2: No

---

## [Author Response · Author response to Decision Letter 0]

27 Jun 2023

Journal Requirements:

3. Please ensure that you include a title page within your main document. You should list all authors and all affiliations as per our author instructions and clearly indicate the corresponding author.

4. Please remove your figures from within your manuscript file, leaving only the individual TIFF/EPS image files, uploaded separately. These will be automatically included in the reviewers’ PDF.

All the requirements above have been addressed (+ coaches consented to participation – Line 141 and the study was reviewed and approved by the Institutional Ethics Board – Line 156).

Specifically for the point 2., we addressed it again with the university ethics’ board and they informed us the parental consent is mandatory for participants under 16 years old and that we are still allowed to use the data from the only 17 years old participant (case by case basis). Please see the line:143-146 within the manuscript and the page 34-35 from the following link: https://www.northumbria.ac.uk/media/27327041/nu-research-ethics-governance-handbook-2016-17.pdf

 The authors would like to thank the reviewers for the valuable comments. We really appreciate your feedback, and we seriously took it into consideration. Please see our comments (and in the new submission the revised manuscript with the track changes):

Reviewer #1: REVIEW:

Reviewer #1: I would like to congratulate the authors on an important piece of work. This manuscript highlights the need for further work in this area and because of that, most of my critical thoughts below are out loud thinking to what is really missing in the literature that can further inform some of the findings here. I recognize the importance of capturing a wide range of participants via the survey mechanism and the importance this brings to the literature, however, the shortcomings of this approach are the lack of clarification/conclusions we can draw based on the quantitative data. Tied to that, my other concern is the lack of athlete-voice. While the authors address this front and center at the beginning of the manuscript, I recommend them to be mindful of this, throughout the discussion where conclusions are drawn.

Most my comments are specific to the text, however, I do urge the authors to revisit the discussion section to see whether some of the conclusive remarks made can be reconsidered given the limited information we have with regards to this cohorts’ development and the need to further unpack some of the nuances here before we can make such conclusions.

Thank you for this comment. We amended the discussion section and presented the conclusions more cautiously. We also highlighted more the need of athlete-voice in research (e.g., limitations and future directions). Please find our amendments within the manuscript and our responses to your comments below. Thank you again for your fruitful comments.

Line 89: close bracket

Thank you for this comment. We corrected it.

Line 101: hinder or impact their progression rate? Reword here, ends awkwardly

Line 80: ‘…could negatively influence their autonomous motivation and in turn, hinder their sports performance progression…’

Line 136: replace trying with ‘aims’ - this is an interesting point you bring up as I was curious to the significance of capturing the athlete/family perspective of this environment, and their experiences

Thank you for your comment. We replaced it.

Line 207: did any of the coaches coach both II and non-II athletes? To this, it might be helpful to get a bit more description of the coaches and sports involved, i.e., years coaching, highest level coaches, level of expertise of athletes they coached (that were rated), sports involved, and cluster of sports, etc.

Thank you for this important comment. It made us consider the importance of providing more information about the sample size:

Line 139: The mean average coaching experience for coaches working with athletes with II was 11 years (SD = 10) while coaches of athletes without II had a mean average coaching experience of 15 years (SD =12). Both groups of coaches had experience coaching a variety of individual and team sports, like fencing, boccia, archery, athletics, football, and basketball. 

Line 144: ...and their athletes were adolescents or adults (aged 12 or above) with or without II.

Line 147: For comparability purposes, their athletes with or without II were categorized in the ‘participation’ or ‘performance’ stage of sports development (focus on sports skills development with experience in local or regional, recreational competitive events)

Line 254-256: I wrestle with this idea and concerned about how it is being interpreted. Isnt rate of progression relative to the athletes’ ability and the coaches’ expectation of their rate of improvement based on their baseline vs growth over time to reach their personal potential? So how is the rate of improvement being measured between the athletes? Did the question ask for rate of improvement relative to the athletes’ expected improvement over that time? And did the athletes’ impairment influence the coaches’ initial expectation standards in any way?

Tied to this, I think relevance of athletes’ level of competition and understanding how that distribution lies among the II and non-II becomes more relevant

Thank you for this important comment. It made us realise the importance of clarifying the athletes' level of sports development and competition:

Line 147: For comparability purposes, their athletes with or without II were categorized in the ‘participation’ or ‘performance’ stage of sports development (focus on sports skills development with experience in local or regional, recreational competitive events)

Provide more clarifications about the rated performance questionnaire (we asked the coaches to act as proxy respondent for a group of persons (athletes with or athletes without II, to be consisted with the approach of the other 2 questionnaires):

Line 159: This instrument is completed by coaches and is used to investigate the extent to which their group of athletes had progressed in the (a) physical, (b) tactical, (c) technical, and (d) psychological domain over the past year [36]. As this is an objective measurement of athletes' perceived sports performance progression, the rate of progression was based on the sports performance abilities of the group of athletes and how coaches perceived their expected rate of improvement.

As there is a possibility the athletes’ impairment influence the coaches’ initial expectation standards, we provided more information about this issue at the discussion section, focussing on the different standards of sport success that coaches of athletes with II may adopt:

Line 273: However, as this study was based on coaches’ perceptions…..

Lastly, we provided at the discussion more information about the relevance of athletes’ level of competition and understanding how that may affect the findings:

Line 287: Moreover, even if we tried to recruit coaches who are working with athletes with a similar stage of sports development, we recognise that disability and non-disability sports organisations are not fully integrated [33]. As a result, the training sessions, the opportunity for sports performance development, and the participation in competitive events may vary for athletes with and without II. Consequently, the coaches' expectations regarding their athletes’ improvement may also differ between the two groups and could partially explain the findings observed in this study [33].

Line 259-265: hard to make this conclusion given the question wasn’t asked of coaches on what they rated as ‘improvement’ and what the metrics were that they considered to analyze athletes’ performance – especially given a range of sports that were involved, ie table-tennis requires a technical proficiency that is more prominent in assessing one’s rate of improvement relative to a cycling where post-balancing, the most relevant measure of improvement is cardiovascular output

We agree with this comment. Thus, we tried to tone it down and give more focus on the coaches' perceptions:

 Line 264: A reason that total performance progression of athletes with II is perceived to be higher could be due to lower long-term engagement in sports and the lower levels of physical fitness and muscle strength of this population compared to athletes without II that previous studies reported [40, 41]. However, due to the nature of the Rated Performance questionnaire (was based on the perceived physical, tactical, technical, and psychological progression of the athletes) and the plethora of different sports that coaches were coaching, we approach this argument with caution. More research based on objective measurements is needed to explore the relationships between training age and physical fitness (e.g., fitness assessments that test the strength and muscle mass alternations of athletes) with the sports performance progression of athletes with II [42, 43].

Line 269-273: more positively and/or the bar is set lower to start with (knowing how many athletes with an impairment the coach has worked with previously would help narrow this assumption down slightly, but these are challenges to surveys vs qualitative studies, harder to draw on coaches’ observations or perception through quantitative measures, so I’d be cautious of the extent that conclusions are drawn from the findings)

We agree with this comment and we amended it accordingly. We also used a recently published qualitative paper with similar aims to strengthen our conclusions. We also provided more information about the coaching experience of the participants.

Line 139: The mean average coaching experience for coaches working with athletes with II was 11 years (SD = 10) while coaches of athletes without II had a mean average coaching experience of 15 years (SD =12).

Line 299: …coaches may have relatively low expectations from their athletes with II…

Line 279: Additionally, coaches of athletes with II tend to adopt a mentorship role, focus less on their athletes’ sports performance development, and potentially underestimate the importance of nurturing the athletic identity that athletes with II may wish to develop [45]

Line 298: and also more research unpacking this directly with the athletes whom has II

This is true:

Line 317: ...and give equal attention to both athletes with II and their coaches [33].

Line 300-302: more context/info on coaches’ experience would help reader make assumptions/draw conclusions on this and other relevant topics

We agree that more info about the coaches' experience will provide more clarity:

Line 139: The mean average coaching experience for coaches working with athletes with II was 11 years (SD = 10) while coaches of athletes without II had a mean average coaching experience of 15 years (SD =12).

Line 320: Given the coaching experience of the participants, with both groups having an average coaching experience of over 10 years, it is unlikely that the observed similarities in coaching styles can be attributed to their (lack of) experience or their experience differences.

Line 320-321: Athletes perhaps respond to the environment that the coach facilitates based on their methods, i.e., they might not directly draw comparisons but experience sense of belongingness, etc. I find this conclusion challenging, as it sounds like athletes explicitly need to deduce coaches’ methods and their preferential against that type of coaching? What if the athlete had never experiences other types of coaching? Is it common for athletes to explicitly identify the coaches’ methods?

Thank you so much for this important comment. We apologise for the confusion, and we paraphrase the sentence as well as we provided more information to enhance the clarity of this conclusion/suggestion:

Line 340: These findings indicate the importance of the coach-athlete relationship in II sports and suggest that athletes with II may have the capability to respond accordingly to different coaching styles contrary to common beliefs [15]. For example, athletes with II may feel a sense of ownership and enjoyment as well as reduced feelings of disinterest when their coaches take time to understand their feelings and needs and provide them with choices and encouragement (need-supportive coaching style) [24]. Additionally, they may feel disengaged, demotivated, and uninterested in participating in sports when their coaches tend to be controlling or neglectful of their needs (need-thwarting coaching style) [24].

Line 322-334: I suggest rewording of some recommendations here, ‘should’ comes off a bit too strong? Especially given how much more work is needed in this area both qualitatively and quantitatively before we make assertions, especially given this was a survey done with coaches and athletes were not directly involved?

Thank you for this comment. We amended it accordingly using a less strong language and we used evidence from a recently published qualitative paper with similar aim.

Limitations: lack of qualitative feedback to the survey info + athlete input is very important acknowledgement, some of the elements you speak of here highlight the need to introduce qualitative work into the quantitative data, highlighting the importance of taking a mixed-methods approach while incorporating the breadth of stakeholders involved in the DTE that impact athletes’ development (i.e., parents, teammates, directors, support staff, coaches, and most importantly, athletes themselves).

Thank you for your comment. We included more information about the limitations and future studies, as you suggested:

Line 391: Another limitation of the study is also the absence of qualitative feedback in the survey data. Qualitative approaches, like interviews with coaches and athletes with II, could provide a deeper understanding of the coach-athlete relationship in disability sports and capture nuanced information that our quantitative approach alone may not revealed [59]. Thus, future studies could use a mixed-methods approach (by combining qualitative and quantitative methods) to obtain a more comprehensive picture of the problem [59]. Future studies should also ensure the active involvement of participants with II and their contribution to the research process. Additionally, it is crucial these studies to also consider other relevant stakeholders (e.g., family members, support staff, policy makers) in examining the coach-athlete relationship in disability sports and the inclusive practices towards athletes with II [45].

Reviewer #2: REVIEW:

In the introduction, in line 99 in this paragraph, the statement "could negatively influence their autonomous motivation and in turn, make them progress less in sports" is only supported by a single article, which is true that it was carried out with a good sample, but it would be important to strengthen this argument with other publications.

Thank you for this comment. We provided two more studies to provide a strengthen the argument (line 82-83).

In the materials and methods in line 159 in relation to the selection of the sample, the criterion of only one year of experience does not seem to be an indicator of being a factor that conveys through your answers a fully founded opinion on the questionnaires that you propose in the study, the lack of experience may be an element that subtracts accuracy from the answers by not being supported by practical knowledge developed through experience.

Thank you for your comment. Even if the lack of experience should not be an issue for these questionnaires, especially as the validity studies recruited coaches with similar coaching experience (e.g., IBQ-self was validated with coaches who had an average of 17.50 years of coaching experience, SD = 12.83), we consider important to provide more information about the sample size:

Line 139: The mean average coaching experience for coaches working with athletes with II was 11 years (SD = 10) while coaches of athletes without II had a mean average coaching experience of 15 years (SD =12). Both groups of coaches had experience coaching a variety of individual and team sports, like fencing, boccia, archery, athletics, football, and basketball.

In the discussion line 264 when you talk about "Thus, our speculation is that the total performance progression of athletes with II might be more obvious to their coaches." I think this part needs more argumentation.

We agree with this comment. Thus, we tried to tone it down and give more important to the coaches' perceptions: 

Line 264: A reason that total performance progression of athletes with II is perceived to be higher could be due to lower long-term engagement in sports and the lower levels of physical fitness and muscle strength of this population compared to athletes without II that previous studies reported [40, 41]. However, due to the nature of the Rated Performance questionnaire (was based on the perceived physical, tactical, technical, and psychological progression of the athletes) and the plethora of different sports that coaches were coaching, we approach this argument with caution. More research based on objective measurements is needed to explore the relationships between training age and physical fitness (e.g., fitness assessments that test the strength and muscle mass alternations of athletes) with the sports performance progression of athletes with II [42, 43].

In line 292 on the reflection "However, the motivational differences between athletes with and without II could have occurred due to the difficulties of proxies (such as coaches) to recognise that people with II can have a good, personally meaningful life [38] and accept the role of people with II in their own autonomous decision-making [32]." It is not clear that the quote [38] is well used here, perhaps I should argue this paragraph more.

Thank you for this comment and apologies for our mistake. 

We updated the citations: 

And provided a more extensive argument, as you suggested:

Line 310: Coaches in sports settings tend to prioritise their own aspirations and perspectives regarding the needs of people with II, potentially overshadowing their athletes' sports motivations [52]. In addition, coaches may observe that the social environment (e.g., parents) hinders the decision-making of people with II and consequently adopt an overprotective stance towards them [52].

---

## [Decision Letter · Decision Letter 1]

8 Dec 2023

Coaching styles and sports motivation in athletes with and without Intellectual Impairments

PONE-D-22-31733R1

Dear Dr. Hettinga,

We’re pleased to inform you that your manuscript has been judged scientifically suitable for publication and will be formally accepted for publication once it meets all outstanding technical requirements.

Kind regards,

Ali B. Mahmoud, Ph.D.

Academic Editor

PLOS ONE

Additional Editor Comments (optional):

Reviewers' comments:

Reviewer's Responses to Questions

**Comments to the Author**

1. If the authors have adequately addressed your comments raised in a previous round of review and you feel that this manuscript is now acceptable for publication, you may indicate that here to bypass the “Comments to the Author” section, enter your conflict of interest statement in the “Confidential to Editor” section, and submit your "Accept" recommendation.

Reviewer #2: All comments have been addressed

2. Is the manuscript technically sound, and do the data support the conclusions?

Reviewer #2: Yes

3. Has the statistical analysis been performed appropriately and rigorously? 

Reviewer #2: Yes

4. Have the authors made all data underlying the findings in their manuscript fully available?

Reviewer #2: Yes

5. Is the manuscript presented in an intelligible fashion and written in standard English?

Reviewer #2: Yes

6. Review Comments to the Author

Reviewer #2: Comments and suggestions have been accepted and adapted by the authors. The article presents the necessary standards to be published in the journal.

7. PLOS authors have the option to publish the peer review history of their article (what does this mean?). If published, this will include your full peer review and any attached files.

Reviewer #2: **Yes: **Garcia-Roca, J.A.

---

## [Editor Report · Acceptance letter]

14 Dec 2023

PONE-D-22-31733R1 

PLOS ONE

Dear Dr. Hettinga, 

I'm pleased to inform you that your manuscript has been deemed suitable for publication in PLOS ONE. Congratulations! Your manuscript is now being handed over to our production team.

Kind regards, 

on behalf of

Dr. Ali B. Mahmoud 

Academic Editor

PLOS ONE